# Neoadjuvant Chemoradiotherapy for Locally Advanced Gastric Cancer: Where Are We at?

**DOI:** 10.3390/cancers14123026

**Published:** 2022-06-20

**Authors:** Jen-Hao Yeh, Yung-Sung Yeh, Hsiang-Lin Tsai, Ching-Wen Huang, Tsung-Kun Chang, Wei-Chih Su, Jaw-Yuan Wang

**Affiliations:** 1Graduate Institute of Clinical Medicine, College of Medicine, Kaohsiung Medical University, Kaohsiung 80708, Taiwan; ed109486@edah.org.tw (J.-H.Y.); tsungkunchang@gmail.com (T.-K.C.); lake0126@yahoo.com.tw (W.-C.S.); 2Division of Gastroenterology and Hepatology, Department of Internal Medicine, E-DA Dachang Hospital, Kaohsiung 82445, Taiwan; 3Department of Medical technology, College of Medicine, I-Shou University, Kaohsiung 82445, Taiwan; 4Division of Gastroenterology and Hepatology, Department of Internal Medicine, E-DA Hospital, Kaohsiung 82445, Taiwan; 5Division of Trauma and Surgical Critical Care, Department of Surgery, Kaohsiung Medical University Hospital, Kaohsiung Medical University, Kaohsiung 80708, Taiwan; 920055@gmail.com; 6Department of Emergency Medicine, Faculty of Post-Baccalaureate Medicine, College of Medicine, Kaohsiung Medical University, Kaohsiung 80708, Taiwan; 7Graduate Institute of Injury Prevention and Control, College of Public Health, Taipei Medical University, Taipei 11031, Taiwan; 8Division of Colorectal Surgery, Department of Surgery, Kaohsiung Medical University Hospital, Kaohsiung Medical University, Kaohsiung 80708, Taiwan; chunpin870132@yahoo.com.tw (H.-L.T.); 1000131@ms.kmuh.org.tw (C.-W.H.); 9Department of Surgery, Faculty of Medicine, College of Medicine, Kaohsiung Medical University, Kaohsiung 80708, Taiwan; 10Department of Surgery, Faculty of Post-Baccalaureate Medicine, College of Medicine, Kaohsiung Medical University, Kaohsiung 80708, Taiwan; 11Center for Cancer Research, Kaohsiung Medical University, Kaohsiung 80708, Taiwan; 12Cohort Research Center, Kaohsiung Medical University, Kaohsiung 80708, Taiwan; 13Pingtung Hospital, Ministry of Health and Welfare, Pingtung 90054, Taiwan

**Keywords:** locally advanced gastric cancer, neoadjuvant treatment, chemoradiation therapy

## Abstract

**Simple Summary:**

More than 50% of gastric cancer are at least locally advanced at presentation. For such patients, a multimodal approach rather than mere surgical resection leads to better long-term prognosis. Neoadjuvant chemoradiotherapy is one of the common treatment strategies for local advanced gastric cancer. Based on the experience and evidence from esophago-gastric cancers, the incorporation of systemic and locoregional therapy has shown superior disease control and reduced local recurrence. However, the optimal chemotherapy regimen, patient selection, technical consideration and potential biomarkers are still under investigation. Furthermore, the comparison of neoadjuvant chemoradiotherapy with neoadjuvant/perioperative chemotherapy is also an important issue to be answered. In the review article, we addressed the current available evidence to provide a comprehensive understanding and the use of neoadjuvant chemoradiotherapy for locally advanced gastric cancer. Future studies and ongoing trials will be necessary to determine the best candidate and the role of newer systemic and radiation therapies in such patients. NCRT is a feasible treatment option for LAGC, with the ability to achieve favorable disease control and enable higher radical resection rates over those afforded by perioperative chemotherapy or surgery alone. Large clinical trials examining the comparative efficacy of NCRT and NCT are underway. The discrepancy between the satisfactory pCR rates associated with NCRT and the nonsignificant association between NCRT and survival warrants further exploration. Furthermore, newer therapies such as immunotherapy and adaptive radiotherapy may be implemented in con-junction with NCRT, and the development of useful biomarkers may ultimately lead to the de-velopment of personalized treatments for LAGC. These research directions may lead to the dis-covery of the optimal approach to administering NCRT to patients with LAGC. They may also aid in the determination of the optimal candidates for undergoing NCRT.

**Abstract:**

Locally advanced gastric cancer (LAGC) has a poor prognosis with surgical resection alone, and neoadjuvant treatment has been recommended to improve surgical and oncological outcomes. Although neoadjuvant chemotherapy has been established to be effective for LAGC, the role of neoadjuvant chemoradiotherapy (NCRT) remains under investigation. Clinical experience and research evidence on esophagogastric junction adenocarcinoma (e.g., cardia gastric cancers) indicate that the likelihood of achieving sustainable local control is higher through NCRT than through resection alone. Furthermore, NCRT also has an acceptable treatment-related toxicity and adverse event profile. In particular, it increases the likelihood of achieving an R0 resection and a pathological complete response (pCR). Moreover, NCRT results in higher overall and recurrence-free survival rates than surgery alone; however, evidence on the survival benefits of NCRT versus neoadjuvant chemotherapy (NCT) remains conflicting. For noncardia gastric cancer, the efficacy of NCRT has mostly been reported in retrospective studies, and several large clinical trials are ongoing. Consequently, NCRT might play a more essential role in unresectable LAGC, for which NCT alone may not be adequate to attain disease control. The continual improvements in systemic treatments, radiotherapy techniques, and emerging biomarkers can also lead to improved personalized therapy for NCRT. To elucidate the contributions of NCRT to gastric cancer treatment in the future, the efficacy, potential toxicity, predictive biomarkers, and clinical considerations for implementing NCRT in different types of LAGC were reviewed.

## 1. Introduction

One of the most common cancers, gastric cancer, constitutes a leading cause of cancer-related death despite improvements in treatment and the widespread eradication of *Helicobacter pylori* [1,2,3]. The suboptimal prognosis of this disease is likely attributable to its aggressive biological behavior and to its frequently advanced stage at diagnosis (in more than 50% of cases) [4]. Although surgical resection provides the highest chance of recovery, it is usually insufficient or inapplicable for locally advanced gastric cancer (LAGC). A multimodal strategy includes systemic and local therapies that are based on the tumor characteristics [5,6]; it can induce disease control, facilitate complete resection, and improve survival outcomes [7]. This principle applies not only to initially resectable disease but also unresectable LAGC [8].

LAGC is typically defined as a tumor of the stomach or esophagogastric junction (EGJ); it is a type of histologically confirmed adenocarcinoma staged under the clinical tumor, node, and metastasis (TNM) staging system as cT3–cT4b, lymph node metastasis (N1–N3) without distant metastases (M0) [9]. In this context, tumors exhibiting mesenteric root invasion, para-aortic lymphadenopathy, or major vessel encasement are considered unresectable. For resectable disease, neoadjuvant chemotherapy (NCT) has demonstrated clear survival benefits over those of initial surgery, regardless of whether adjuvant chemotherapy was implemented [10,11,12]. Moreover, NCT might result in the downstaging of LAGC, facilitating subsequent resection [13,14,15]. Little information is available on the addition of radiotherapy, namely, neoadjuvant chemoradiotherapy (NCRT), to LAGC treatment programs.

A network meta-analysis concluded that combining radiotherapy and chemotherapy leads to more favorable local control relative to modality alone [16]. According to clinical trials on esophageal or EGJ adenocarcinomas, NCRT is associated with a significantly lower local failure rate and higher pathological complete response (pCR) and R0 resection rates in subsequent surgery [17,18,19]. Furthermore, NCRT results in a more satisfactory clinical response than NCT, suggesting its viability as a treatment modality. Prognostic data for LAGC are less abundant than those for esophageal and EGJ cancers. Notably, several clinical trials exploring the efficacy and safety of NCRT in LAGC are ongoing [20,21,22,23]. This paper presents a narrative review of studies on NCRT for LAGC, examining its efficacy, adverse effects, technical aspects, and perioperative and oncological outcomes. Furthermore, we discuss recommendations for the implementation of NCRT in this context.

## 2. Material and Methods

We searched the PubMed and Cochrane Library databases in February 2022 for English-language articles. The search strategy is described in Appendix A. In brief, the titles and abstracts of the publications retrieved from the aforementioned databases were screened. Subsequently, relevant articles were manually reviewed to identify other potentially eligible studies.

## 3. NCRT for EGJ and Gastric Cardia Cancers

### 3.1. NCRT versus Surgery Alone

Multimodal treatment has been advocated for locally advanced EGJ and esophageal cancer because of the poor survival rate afforded by radical surgery alone [24,25,26]. Specifically, NCRT or perioperative chemotherapy is recommended for EGJ adenocarcinoma [27]. EGJ adenocarcinoma can be further classified as esophageal or gastric cancer, with a distinct staging system for each type of cancer under a staging system slightly different from the TNM staging system. In general, EGJ tumors are staged as gastric cancer if they extend more than 2 cm to the proximal stomach; otherwise, they are staged as esophageal cancer [28,29]. Under the Siewert classification, which is widely applied to the classification of EGJ cancers, type I and type II/III tumors are more appropriately staged as esophageal and gastric cancer, respectively [30]. However, gastric cardia cancers are frequently included with EGJ adenocarcinoma in clinical studies, and their management is largely the same.

The superior survival benefits conferred by NCRT over surgery alone for locally advanced esophageal cancer and EGJ cancer were first demonstrated in an Irish clinical trial in which 113 patients with esophageal adenocarcinoma were randomly assigned to receive either NCRT or surgery alone. The 3 year overall survival (OS) rate achieved through NCRT was significantly higher than that achieved through surgery (32% vs. 6%, *p* = 0.01) [31]. In another trial, CALGB 9781, in which patients with esophageal adenocarcinoma constituted the majority, NCRT also resulted in more favorable survival over surgery alone (median OS, 4.48 vs. 1.79 years, *p* = 0.02) [32]. Moreover, in the phase III CROSS trial [33], NCRT was associated with a higher R0 resection rate (92% vs. 69%, *p* < 0.001) and OS (hazard ratio [HR] 0.65, 95% confidence interval [CI] 0.49–0.87) than surgery alone. The rate of major adverse events associated with NCRT was acceptable (6% leukopenia and 5% anorexia), and in-hospital mortality did not differ between the two groups. Furthermore, the survival benefits afforded by NCRT persisted over 10 years in the long-term follow-up [34]. Although both squamous cell cancer and adenocarcinoma were considered in the trial, 75% of the patients had adenocarcinoma, and NCRT led to survival benefits in both types of cancer. 

Other trials have reported negative results for NCRT. Aside from two studies that were underpowered due to the low number of cases [35,36], the FFCD 9901 trial, which included patients with stage I and II esophageal cancer, found that NCRT did not provide any survival benefits over surgery alone. Instead, it reported a significantly higher postoperative mortality rate of 11.1% of NCRT versus 3.4% of NCT (*p* = 0.049) [37]. These discrepant findings may be explained by between-study differences in patient characteristics; only 29.2% of the patients had adenocarcinoma, and most tumors were located at the middle-third of the esophagus. Although subgroup analysis for stage I and II tumors was not performed, the present study indicated that NCRT should be considered with caution for earlier stage disease. On the other hand, meta-analyses have consistently indicated that NCRT confers greater survival benefits than surgery alone for locally advanced esophageal and EGJ adenocarcinoma [38,39,40], and that these benefits may be more pronounced in younger patients (patients aged ≤55 years) [39]. The clinical studies discussed thus far are summarized in Table 1.

### 3.2. NCRT versus NCT for EGJ and Gastric Cardia Cancers

Since the MAGIC trial reported that perioperative chemotherapy with ECF regimen (i.e., epirubicin, cisplatin, and fluorouracil) resulted in a significantly more favorable clinical response and significantly higher OS over surgery alone for distal esophageal and gastric cardia adenocarcinoma [10], researchers have devoted efforts to determining whether NCRT or NCT is more suitable for gastric cardia cancers. The German POET trial is the only randomized controlled Phase III trial designed for EGJ cancer that compares NCRT and NCT [19,41]. Patients undergoing NCRT had a higher rate of local recurrence-free survival (RFS; HR 0.37, 95% CI 0.16–0.85) as well as a higher rate of pCR (14.3% vs. 1.9%, *p* = 0.03) and a trend toward higher 5 year OS (39.5% vs. 24.4%, HR 0.65, 95% CI 0.42–1.01). Notably, the subgroup analysis suggested that patients with cardia cancers (Siewert type II) gained more benefits from NCRT relative to patients with Siewert type I cancers. 

Conversely, the phase III NEO-AEGIS trial [42] and a Swedish trial [18] indicated that NCRT did not confer greater benefits in terms of OS and RFS than NCT, despite the association of NRT with higher pCR and R0 resection rates. Moreover, a meta-analysis suggested that NCRT is associated with higher postoperative mortality rates than is NCT (relative risk (RR) 1.58, 95% CI 1.00–2.49) [16]. In summary, evidence from locally advanced EGJ cancer indicates that NCRT is the modality of choice in terms of local control, although whether it affords greater survival benefits over NCT remains unclear. Until more evidence from clinical trials is presented, the implementation of NCRT in cases of gastric cardia cancer can be considered [19,42,43]. 

Recently, the results of the recent phase II/III FLOT4 trial [44] suggest a new standard for perioperative chemotherapy for EGJ cancers and LAGC. The perioperative FLOT regimen, which comprises fluorouracil plus leucovorin, oxaliplatin, and docetaxel, provided superior OS (median, 50 vs. 35 months, HR: 0.77, 95% CI 0.63–0.94) relative to the ECF or ECX (i.e., epirubicin, cisplatin, and capecitabine) regimens. Although numerous patients may benefit from perioperative FLOT, whether it can be a substitute for NCRT remains unclear [44], and a clearer answer may emerge after the completion of the ESOPEC trial, which directly compares the perioperative FLOT and CROSS regimens.

## 4. NCRT for Locally Advanced, Resectable Noncardia Gastric Cancer

Based on the experience of and evidence from research on EGJ and cardia cancers, the main advantage of NCRT is that it achieves a higher rate of local control to enable subsequent curative surgery. Compared with its use in EGJ and cardia cancers, the use of NCRT for noncardia gastric cancer is less validated due to the lack of phase III randomized controlled trials. Evidence from mostly uncontrolled studies [13,45,46,47,48,49,50,51,52] indicates that NCRT led to R0 resection and pCR rates of approximately 70–80% and approximately 20–25%, respectively. A review of the performances of NCRT and other modalities is presented as follows. 

### 4.1. NCRT versus Adjuvant Therapy for Resectable LAGC

A small trial found that NCRT afforded no clinical benefits over adjuvant chemoradiotherapy [53]. However, two recent studies with propensity score matching suggested that NCRT is preferred over adjuvant chemotherapy [54] or chemoradiotherapy [55]. In a Chinese cohort, NCRT was associated with a significantly higher pCR rate (17.0% vs. 4.0%, *p* = 0.001), RFS (HR, 0.63; 95% CI 0.43–0.92, *p* = 0.014), and local-recurrence-free survival rates (HR, 0.40; 95% CI 0.23–0.69, *p* = 0.0019) but a significantly higher proportion of grade 3/4 adverse events (52% vs. 34%, *p* = 0.01). The OS did not differ significantly between treatments (HR, 0.45; 95% CI 0.51–1.11, *p* = 0.15) [54]. In contrast, in a Korean cohort, NCRT was associated with significantly improved OS (HR 0.57, 95% CI 0.36–0.91, *p* = 0.020) and R0 resection rates (HR 0.50, 95% CI 0.27–0.90, *p* = 0.021) as well as lower grade 3/4 toxicity (10% vs. 54%, *p* < 0.001) than adjuvant chemoradiotherapy [55].

A recent randomized controlled trial examined adjuvant XELOX chemotherapy administered to 60 patients with LAGC and compared the outcomes of adjuvant XELOX chemotherapy with and without NCRT [56]. NCRT resulted in a significantly higher RFS rate (60.0% vs. 33.3%, *p* = 0.019) and a significantly lower local recurrence rate (11.5% vs. 36.7%, *p* = 0.039) for up to 3 years, without an increase in perioperative complications (23.1% vs. 30.0%, *p* = 0.560). No significant difference in OS was observed (63.3% vs. 50.0%, *p* = 0.215). These findings, which are summarized in Table 2, indicate that NCRT is more effective than adjuvant treatments in achieving and maintaining local control. To determine whether long-term OS can be improved under NCRT, further investigations are warranted.

### 4.2. NCRT versus NCT for Resectable LAGC

NCT has demonstrated survival benefits over upfront surgery [10,57], and this approach is embraced under current treatment guidelines [8]. Nonrandomized studies have compared NCRT and NCT (Table 3) [46,51,52,54,58,59]. In general, NCRT is more likely to achieve a pCR and enable R0 resection than is NCT. However, the advantage of more favorable local control does not confer OS benefits. Trumbull et al. [59] observed greater survival benefits in terms of pCR under NCT than under NCRT in patients with LAGC. Another study found that NCRT was beneficial in patients with initial lymph node metastasis, as indicated by the association of a complete nodal response with significantly improved survival. Martin-Romano et al. reported that compared with patients receiving NCT, patients receiving NCRT had a higher likelihood of achieving a better local response (Becker Ia-b response, 58 vs. 32%, *p* = 0.001), exhibiting a grade D nodal regression (30% vs. 6%, *p* = 0.009), and attaining a favorable pathological response (23% vs. 3%; *p* = 0.019). However, no between-group difference in survival in patients with no cancerous lymph nodes at baseline was detected [51].

Reasons explaining why more favorable disease control does not necessarily translate to longer survival include insufficient follow-up durations, differences in chemotherapeutics, selection bias ascribable to a retrospective design, and potential toxicity attributable to radiotherapy. Furthermore, whether the NCRT outcomes vary with the location of the tumor remains unclear. A study based on the US National Cancer Database found that NCRT resulted in less favorable outcomes than perioperative chemotherapy (HR 1.10, 95% CI 1.00–1.20) in proximal gastric cancer. Corresponding data for distal gastric cancer are lacking [60]. Notably, NCRT remains recommended as an alternative to NCT in the guidelines of the US National Comprehensive Cancer Network. At least two large randomized controlled trials (i.e., TOPGEAR and CRITICS-II) evaluating the feasibility of NCRT for resectable LAGC are ongoing [20,61].

### 4.3. NCRT versus NCT for Unresectable LAGC

Less uncertainty surrounds the optimal approach for unresectable LAGC. In general, treatment should be administered to improve survival rates and quality of life and to palliate symptoms. For medically fit patients, systemic therapies are frequently recommended. Among various modalities, NCRT has been reported to result in a wide-ranging response rate (33–83%, Table 4) [62,63,64,65,66], and under the exclusion of metastatic disease, a clinical complete response was observed in 23–36% of patients [63,64]. In a phase II study including 29 patients, the initial NCRT response rate was 55%, and no clinical complete response was achieved. However, R0 resection was attained in 10 patients undergoing subsequent surgical resection, and pCR was achieved in 4 of those individuals [62].

An observational study was conducted to compare NCRT and NCT in unresectable and metastatic LAGC. Compared with NCT, NCRT resulted in significantly longer OS (median 14 months vs. 10 months, *p* = 0.011) and RFS (median 9 months vs. 8 months, *p* = 0.008). The occurrence of grade 3/4 toxicity did not differ significantly between the two groups [65]. A recent retrospective study determined that in patients with LAGC, compared with NCT, neoadjuvant chemotherapy was associated with a more favorable pathologic response, with no increase in serious postoperative complications [67]. Furthermore, an investigation based on the US National Cancer Database found that in cases of in unresectable LAGC, NCRT was associated with higher survival rates relative to chemotherapy alone. Only 30.8% of patients received NCRT in that study, suggesting the potential underutilization of this modality [66].

Taken together, the evidence indicates that NCRT is a viable option for patients with unresectable LAGC with favorable performance status. However, further randomized controlled trials are warranted to explore the optimal approach to treating this patient population.

## 5. Selection of Chemotherapy Regimen of NCRT for LAGC

The optimal chemotherapy regimen for NCRT has yet to established. The primary regimen used in clinical trials is doublet chemotherapy; the types of doublet chemotherapy include cisplatin plus 5-fluorouracil [32], capecitabine plus oxaliplatin [68], and the CROSS regimen, which comprises low-dose carboplatin plus paclitaxel [33] and is recommended by the latest NCCN guidelines [8]. Although a direct comparison of the aforementioned treatments has yet to be conducted, the CROSS regimen is suggested by various experts because of its ease of administration (single weekly dose) and favorable safe profile, especially for mucositis. Furthermore, a recent network meta-analysis of esophageal cancer also suggested that the CROSS regimen is preferred over a 5-fluorouracil-based regimen [69]. In addition, whether induction chemotherapy followed by chemoradiation can improve the efficacy of NCRT remains unclear [58,70], and this topic may be further clarified by the ongoing CRITICS-II trial [61].

## 6. Toxicity, Therapeutical Considerations, and Biomarkers of NCRT

NCRT is generally considered well tolerated, and overall complication rates have been noted to be comparable to those of NCT [46,65,67,71] and surgery alone [71,72,73]. However, a study based on a US national database suggested that NCRT is associated with significantly higher 30 day postoperative mortality rates (2.91% vs. 1.68% and 0.04%, *p* < 0.001) and significantly higher 90 day postoperative mortality rates (7.09% vs. 4.63% and 0.39%, *p* < 0.001) than perioperative chemotherapy or adjuvant chemoradiotherapy. In that study, NCRT was primarily administered to patients with proximal LAGC. This finding may partly explain why NCRT did not produce significant survival benefits despite providing more favorable local disease control in other studies [46,51,52,54,58,59]. However, this supposition necessitates validation through further large-scale clinical trials. In addition, Fujitani et al. suggested that an age of >60 years (RR: 11.3, 95% CI: 2.50–50.6) and a body mass index of ≥26 (RR: 4.08, 95% CI: 1.08–15.4) are significant risk factors for postoperative complications in patients who underwent NCRT following induction chemotherapy [74].

Radiotherapy, which is a crucial component of NCRT, has evolved as a medical treatment [75]. In clinical practice, radiotherapy is administered on the basis of tumor location and node metastasis status. For proximal gastric cancer, the field must cover the splenic hilum and paraesophageal lymph nodes; as for distal gastric cancer, coverage should encompass the porta hepatis, the distal superior mesenteric artery, and the duodenal stump [76]. The use of radiotherapy was evaluated because the earlier INT0116 trial suggested successful local control and a potential survival benefit with adjuvant chemoradiotherapy compared with surgery alone, even when old parallel-opposed anterior and posterior field arrangements were implemented [77]. Subsequently, a study demonstrated that the newer three-dimensional confocal radiotherapy can be safely administered with the more effective ECF chemotherapy regimen [78], and it has become the backbone of the TOPGEAR trial [20]. Notably, the role of adjuvant chemoradiotherapy has decreased because of its lack of advantages versus perioperative chemotherapy [21,79], and ongoing studies are focusing on the role of NCRT.

Modern radiotherapy includes dynamic planning techniques (e.g., image-guided radiotherapy, adaptive radiotherapy [80], and four-dimensional computed tomography [81]) because of the mobility of the stomach. Furthermore, intensity-modulated radiotherapy (IMRT) may reduce extra-organ damage to healthy tissues adjacent to the treatment site better than conventional three-dimensional conformal radiotherapy [82,83]. The combination of newer planning and therapeutic modalities can minimize the radiation dose and associated toxicity. Adaptive radiotherapy, which uses deformation models to adapt to individual gastric shapes, has been validated, and a population-based model is available [84]. The progress achieved can improve the safety and effectiveness of the treatment in a personalized manner.

Several biomarkers have been associated with the outcomes of NCRT-treated LAGC, including overexpressed ERCC1 and ERCC2 [4], LAG-3 [85], microRNAs (miRs) 338-3p and miR-142-3p [86], T-cell density, and 5-fluorouracil-related enzymes [87,88]. Before these biomarkers are applied to clinical practice, further validation of these findings is required. Notably, the presence of signet ring cell carcinoma and a higher histologic grade may be associated with chemoradiation resistance and poorer prognosis [89]. Shared decision making and participation in clinical trials should thus be considered for such patients whenever possible.

## 7. Conclusions and Future Perspectives

NCRT is a feasible treatment option for LAGC, with the ability to achieve favorable disease control and enable higher radical resection rates over those afforded by perioperative chemotherapy or surgery alone. Large clinical trials examining the comparative efficacy of NCRT and NCT are underway. The discrepancy between the satisfactory pCR rates associated with NCRT and the nonsignificant association between NCRT and survival warrants further exploration. Furthermore, newer therapies such as immunotherapy and adaptive radiotherapy may be implemented in conjunction with NCRT, and the development of useful biomarkers may ultimately lead to the development of personalized treatments for LAGC. These research directions may lead to the discovery of the optimal approach to administering NCRT to patients with LAGC. They may also aid in the determination of the optimal candidates for undergoing NCRT.

## Figures and Tables

**Table 1 cancers-14-03026-t001:** Studies examining neoadjuvant chemoradiotherapy (NCRT) for esophagogastric junction (EGJ) cancer or gastric cardia cancer (GCC).

Author	Trial Name	Patients	Group	Chemotherapy	Radiotherapy	R0 Resection of NCRT (%)	pCR of NCRT (%)	Survival Outcomes
Walsh et al., 1996 [31]		113 EGJ AC	NCRT vs. surgery	PF × 2 4-weekly	40 Gy, 2D/3D EBRT	92.9	25	3 year OS rate was higher under NCRT vs. surgery alone (32% vs. 6%, *p* = 0.01).
Tepper et al., 2008 [32]	CALGB-9781	56 EC (75% EGJ AC)	NCRT vs. surgery	PF × 2 monthly	50.4 Gy, EBRT	NA	40	Median OS was 4.48 years vs. 1.79 years, favoring NCRT (*p* = 0.002).
van Hagen et al., 2012 [33]	CROSS	366 EC (75% EGJ AC)	NCRT vs. surgery	CP × 5 every week	41.4 Gy, 3D EBRT	92	29	Median OS was 49.4 months vs. 24.0 months, favoring NCRT (*p* = 0.003).
Urba et al., 2001 [35]		100 EC (75% EGJ AC)	NCRT vs. surgery	PF × 2 + vinblastine	45 Gy, 3D EBRT	NA	28	Median OS was 17.6 months with surgery alone vs. 16.9 months with NCRT. (*p* = 0.15).
Burmeister et al., 2005 [36]		128 EC (62% EGJ AC)	NCRT vs. surgery	PF × 1	35 Gy, 2D EBRT	80	NA	Similar OS (HR: 0.89, 95% CI: 0.67–1.19) and RFS (HR 0·82, 95% CI 0.61–1.10) were observed between NCRT and surgery.
Mariette et al., 2014 [37]	FFCD-9901	195 EC (28% EGJ AC) Stage I-II	NCRT vs. surgery	PF × 2 biweekly	45 Gy, 3D EBRT	93.8	33.3	NCRT had a similar 3 year OS rate (47.5% vs. 53.0%, *p* = 0.94) but a higher postoperative mortality rate (11.1% vs. 3.4%, *p* = 0.049).
Stahl et al., 2017 [19,41]	POET	126 Pts (EGJ AC/GCC)	NCRT vs. NCT	*NCRT*: Induction PLF × 2 then PE *NCT*: PLF × 2.5	30 Gy, 3D EBRT	69.5	15.6	NCRT had a similar 5 year OS rate (39.5% vs. 24.4%, *p* = 0.055) but higher local RFS (HR: 0.37, 95% CI 0.16–0.85) vs. NCT.
Reynold et al., 2021 [42]	Neo-AEGIS	377 Pts (EGJ or Esophageal AC)	NCRT vs. NCT	*NCRT*: CP × 5 every week *NCT*: FLOT	41.4 Gy 3D/4D EBRT	95	16	3 year OS rate was similar (56% with NCRT vs. 57% with NCT, HR: 1.02, 95% CI: 0.74–1.42, *p*-value was not available).
Tsai et al., 2020 [43]		5,371 GCC	NCRT vs. NCT	NA (US national database)	NA	91.4	NA	Multivariable analysis revealed similar OS (HR 0.95, 95% CI 0.86–1.05).
Klevebro et al., 2016 [18]		181 Pts (72% EGJ/28% Esophageal AC)	NCRT vs. NCT	*NCRT*: PF × 3 every 3 week *NCT*: PF × 3	40 Gy, 3D EBRT	87	28	3 year OS rate was similar (47% with NCRT vs. 49% with NCT, *p* = 0.77). RFS was 44% in both groups.

AC: adenocarcinoma; EC: esophageal cancer; pCR: pathological complete response; OS: overall survival; RFS: recurrence-free survival; PF: cisplatin plus fluorouracil; CP: carboplatin plus paclitaxel; EBRT: external beam radiation therapy; PLF: cisplatin, leucovorin, and fluorouracil; PE: cisplatin and etoposide; FLOT: fluorouracil plus leucovorin, oxaliplatin, and docetaxel; HR: hazard ratio; CI: confidence interval; US: United States; NCT: neoadjuvant chemotherapy; NA: not available.

**Table 2 cancers-14-03026-t002:** Studies examining neoadjuvant chemoradiotherapy (NCRT) for locally advanced gastric cancers (LAGC) in comparison with surgery alone or adjuvant therapies.

Author	Trial Name	Patients	Group	Chemotherapy	Radiotherapy	R0 Resection of NCRT (%)	pCR ofNCRT (%)	Survival Outcomes
Ajani et al., 2006 [13]	RTOG- 9904	43 NCGC	NCRT	Induction PF × 1 then cisplatin + paclitaxel	45 Gy, 3D EBRT	77	26	Median OS was 23.2 months. R0 resection and pCR were associated with improved outcomes (*p*-value not shown).
Ajani et al., 2004 [45]		33 NCGC (all resectable)	NCRT	Induction PF × 1 then fluorouracil	45 Gy, 2D EBRT	70	30	Median OS was 33.7 months.
Pepek et al., 2013 [47]		48 GC (73% proximal)	NCRT	Various	45 Gy, 3D EBRT	86	19	3 year OS and RFS rates were 50% and 41%, respectively.
Rostom et al., 2013 [48]		41 GC/EGJ AC (68% NCGC)	NCRT	Induction PF × 2 then fluorouracil	45 Gy, 3D EBRT	70.7	24	3 year OS rate was 47.3%. R0 resection (*p* = 0.027) and pCR (*p* = 0.01) were associated with improved outcomes.
Trip et al., 2014 [49]		24 NCGC	NCRT	Carboplatin plus paclitaxel × 5	45 Gy, 3D IMRT	72	16	Median OS was 15 months.
Badgwell et al., 2015 [50]		192 (74% GC)	NCRT	NA	NA	93	20	5 year OS was 56% (median OS: 5.8 years).
Saedi et al., 2014 [53]		25 NCGC	NCRT vs. Surgery	PF × 1 then Adjuvant ECX	45 Gy, 2D EBRT	NA	NA	5 year OS rates were similar (38.5% with NCRT vs. 16.7% with surgery, *p* = 0.169).
Kim et al., 2022 [55]		152 GC/EGJ AC (42% NCGC)	NCRT vs. ACRT	Various	50.4 Gy, IMRT	95	26	NCRT was independently associated with improved OS (HR: 0.57, 95% CI: 0.36–0.91).
Wang et al., 2021 [56]		60 NCGC	NCRT vs. ACT	XELOX × 2	50.4 Gy, 3D EBRT	84.6	NA	3 year OS rates were similar (60% with NCRT vs. 50% with ACT, *p* = 0.215).

NCGC: noncardia gastric cancer; EGJ: esophagogastric junction; AC: adenocarcinoma; pCR: pathological complete response; OS: overall survival; RFS: recurrence-free survival; PF: cisplatin plus fluorouracil; EBRT: external beam radiation therapy; IMRT: intensity modulated radiation therapy; ECX: epirubicin, cisplatin, and capecitabin; XELOX: oxaliplatin plus capecitabine; NCT: neoadjuvant chemotherapy; ACT: adjuvant chemotherapy; ACRT: adjuvant chemoradiotherapy; HR: hazard ratio; CI: confidence interval.

**Table 3 cancers-14-03026-t003:** Studies examining neoadjuvant chemoradiotherapy (NCRT) for locally advanced gastric cancer (LAGC).

Author	Trial Name	Patients	Group	Chemotherapy	Radiotherapy	R0 Resection of NCRT (%)	pCR ofNCRT (%)	Survival Outcomes
An et al., 2013 [46]		74 NCGC (all resected)	NCRT vs. NCT	Various	45 Gy, mode not shown	87.8 (combined)	NA	OS was similar between NCRT and NCT (*p*-value not shown).
Martin-Romano et al., 2016 [51]		80 NCGC	NCRT vs. NCT	NA	45 Gy, 3D EBRT	95.3	23.3	Median OS was similar (71 months with NCRT vs. 51 months with NCR, *p* = 0.24).
Zhang et al., 2016 [52]		126 GC	NCRT vs. NCT	*NCRT*: NA *NCT*: docetaxel and S-1	NA	89.7	15.5	3 year OS rates were similar (46.6% vs. 37.0%, *p*-value not shown).
Wang et al., 2021 [54]		2779 GC	NC(R)T vs. ACT	*NCRT*: SOX × 2–4 *NCT/ACT*: various	45 Gy, IMRT	86	17	NCRT was associated with longer OS relative to ACT (52 months vs. 26 months, *p* < 0.001). OS results for NCRT/NCT were similar.
Allen et al., 2021 [58]		440 GC	NCRT vs. NCT	Induction: various, then fluorouracil for NCRT	45 Gy, IMRT	NA	27.7	Median OS was borderline longer with NCRT (122.1 vs. 70.7 months, *p* = 0.21).
Trumbull et al., 2021 [59]		413 GC	NCRT vs. NCT	NA (US national database)	NA	NA	100%	Only patients with PCR were enrolled. NCRT had worse 5 year OS rates relative to NCT (60% vs. 94%, *p* < 0.001).
Barzi et al., 2020 [60]		35,882 GC	NCRT vs. NCT vs. others	NA (US national database)	NA	NA	NA	For proximal GC, NCRT was inferior to PCT (HR: 1.1, 95% CI: 1.00–1.20). No data were reported for distal GC with NCRT.
Leong et al., 2017 [20]	TOPGEAR	120 GC	NCRT vs. PCT	NCRT: ECF induction, then fluorouracil PCT: ECF × 3	45 Gy, IMRT	NA	NA	An interim analysis indicated that 90% and 85% of patients receiving PCT and NCRT, respectively, proceeded to surgery. Grade 3+ toxicity was 22% in both groups.

NCGC: noncardia gastric cancer; GC: gastric cancer; pCR: pathological complete response; OS: overall survival; SOX: S-1 and oxaliplatin; EBRT: external beam radiation therapy; IMRT: intensity modulated radiation therapy; NCT: neoadjuvant chemotherapy; PCT: perioperative chemotherapy; ECF: epirubicin, cisplatin, and 5-fluorouracil; HR: hazard ratio; US: United States; CI: confidence interval.

**Table 4 cancers-14-03026-t004:** Studies examining neoadjuvant chemoradiotherapy (NCRT) for unresectable gastric cancer (GC).

Author	Trial Name	Patients	Group	Chemotherapy	Radiotherapy	R0 Resection of NCRT (%)	pCR of NCRT (%)	Survival Outcomes
Saikawa et al., 2008 [62]		30 GC	NCRT	S-1 with low dose cisplatin	40 Gy, 2D EBRT	100 (33.3% received surgery)	13	Median OS was 25 (range: 10–50) months.
Liu et al., 2017 [63]		36 GC	NCRT	Modified DCF before and after RT; docetaxel with RT	50.4 Gy, IMRT	NA	NA	Median survival time was 25.8 months (95% CI: 7.1–44.5 months).
Taki et al., 2017 [64]		21 GC	NCRT	Various	50 Gy, 3D EBRT	NA	NA	Clinical complete response rate was 16.6%, and the mean OS was 19.8 (range: 3–51) months.
Yeh et al., 2020 [65]		65 GC (46% NCRT)	NCRT vs. NCT	mFOLFOX-4	45-50 Gy, 3D EBRT and IMRT	36.7	NA	NCRT had higher median OS (14 vs. 10 months, *p* = 0.011) and RFS (9 vs. 8 months, *p* = 0.008) relative to NCT.
Li et al., 2018 [66]		4795 GC	NCRT vs. NCT	NA (US national database)	45 Gy (median)	NA	NA	Multivariable analysis and propensity score matching revealed that NCRT was associated with improved OS (HR: 0.82, 95% CI: 0.77–0.89) relative to NCT.

GC: gastric cancer; pCR: pathological complete response; OS: overall survival; RFS: recurrence-free survival; RT: radiation therapy; EBRT: external beam radiation therapy; IMRT: intensity modulated radiation therapy; NCT: neoadjuvant chemotherapy; PCT: perioperative chemotherapy; DCF: doxorubicin, cisplatin, and 5-fluorouracil; FOLFOX: fluorouracil, leucovorin, and oxaliplatin; US: United States; HR: hazard ratio; CI: confidence interval.

## Data Availability

All data were directly extracted or appropriately calculated based on the cited articles.

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
