# Peer review of "Neoadjuvant Chemoradiotherapy for Locally Advanced Gastric Cancer: Where Are We at?"

_cancers, 2022, doi:10.3390/cancers14123026_

Round 1

Reviewer 1 Report

The authors reviewed in this study the role of neoadjuvant chemoradiotherapy for locally advanced gastric cancer.

First of all, the article is well written, the presentation is clear and the review is comprehensive regarding the published evidence on NRCT for gastric cancer (Vs surgery alone, NCT alone or adjuvant treatment). However, some points need to be addressed before publication.

Major revisions:

The article lacks a chapter on the technological evolution of radiation therapy for gastric and esophagogastric junction cancers and its role in interpreting published results. Paragraph 5 is insufficient to fulfill this role. As this is a review focusing on the role of radiotherapy, a review of the current strategy of technological development in this field seems important  in order to put into perspective the potential contribution of radiotherapy in the coming years. The role of IMRT, IGRT and 4D and multimodal planning should be discussed. A “future directions” section, including adaptive radiotherapy and personalized radiotherapy might bring an added value to the article

Additionally, I would recommend to report the delivered doses of radiotherapy and the techniques used for delivery (IMRT, 3D CRT or 2D CRT; and Image guided radiotherapy (IGRT) (yes/no)) in the tables.

Furthermore, the role of concomitant chemotherapy and the type of chemotherapy used should be discussed in a specific paragraph. The type of chemotherapy used should be reported in the tables (drugs and number of cycles)

Minor revisions:

I would recommend to revise the tables of the manuscript to make the reading easier and clearer. The presentation is quite difficult to read and the “Outcomes and notes” section provide information that is not well separated between the different studies.

Some abbreviations are not defined at the time of appearance in the article: NRCS (table 1)

Some statement are inaccurate and should be modified:

“Furthermore, the radiation dose (45 Gy in 25 fractions) was higher than that in the CROSS trial (41.4 Gy in 23 fractions). This higher dose may have induced higher rates of radiation pneumonitis and impaired pulmonary function.”

It is very unlikely that such a small difference in dose could induce the higher rates of toxicity. Other explanations must be sought, like the different chemotherapy regimen used or differences in patient characteristics.

“Regarding radiotherapy, dynamic modalities such as intensity-modulated radiotherapy (IMRT) might be preferable because of their ability to comply the mobility of stomach”

This sentence is not correct. The authors might refer to Image guided radiotherapy (IGRT), adaptive radiotherapy and/or 4D CT for radiotherapy planning. These techniques enable to take into account the mobility of stomach. On the contrary, IMRT does not take into account this mobility but enables to improve radiotherapy dosimetry with better planning target coverage and organs at risk sparing.

In conclusion, the article is interesting and well written but needs a review of the technical aspects and the latest technological developments in radiotherapy.

Author Response

We appreciate your diligent review and precious comments to improve this article. Our responses and revision are presented below according to the comments:

  1. The article lacks a chapter on the technological evolution of radiation therapy for gastric and esophagogastric junction cancers and its role in interpreting published results. Paragraph 5 is insufficient to fulfill this role. As this is a review focusing on the role of radiotherapy, a review of the current strategy of technological development in this field seems important in order to put into perspective the potential contribution of radiotherapy in the coming years. The role of IMRT, IGRT and 4D and multimodal planning should be discussed. A “future directions” section, including adaptive radiotherapy and personalized radiotherapy might bring an added value to the article

Response:

Thanks for your comments, we have refined and supplemented the statements in the Toxicity, therapeutical considerations, and biomarkers of NCRT (Page 14, line 5-26) section, and specifically focused the development of radiation therapy technique. We have also revised the statements in the Conclusion and Future Perspectives (Page 15, line 18-20), and hope the revision meet your standard for this article.

  1. Additionally, I would recommend reporting the delivered doses of radiotherapy and the techniques used for delivery (IMRT, 3D CRT or 2D CRT; and Image guided radiotherapy (IGRT) (yes/no)) in the tables.

Response:

Thanks for your comments. We have added this information in Table 1-4 accordingly.

  1. Furthermore, the role of concomitant chemotherapy and the type of chemotherapy used should be discussed in a specific paragraph. The type of chemotherapy used should be reported in the tables (drugs and number of cycles)

Response:

Thanks for your comments. We have added a new paragraph in the section Selection of chemotherapy regimen of NCRT for LAGC focusing on the choice of chemotherapy regimen of NCRT. (Page 13, line 1-14) We also have added the information of chemotherapy of each study in Table 1-4 accordingly.

  1. I would recommend revising the tables of the manuscript to make the reading easier and clearer. The presentation is quite difficult to read, and the “Outcomes and notes” section provide information that is not well separated between the different studies.

Response:

Thanks for your comments. All the Tables (1-4) are revised to be more concise according to your recommendation.

  1. Some abbreviations are not defined at the time of appearance in the article: NRCS (table 1)

Response:

Thanks for your kind reminder. We have added the abbreviation and its meaning in the legend of Table 1.

  1. Some statement are inaccurate and should be modified:“Furthermore, the radiation dose (45 Gy in 25 fractions) was higher than that in the CROSS trial (41.4 Gy in 23 fractions). This higher dose may have induced higher rates of radiation pneumonitis and impaired pulmonary function.” It is very unlikely that such a small difference in dose could induce the higher rates of toxicity. Other explanations must be sought, like the different chemotherapy regimen used or differences in patient characteristics.

Response:

Thanks for your comments. We have modified the statements here in the NCRT for EGJ and gastric cardia cancers section (Page 7, line 26 to page 8, line 1-4).

  1. “Regarding radiotherapy, dynamic modalities such as intensity-modulated radiotherapy (IMRT) might be preferable because of their ability to comply the mobility of stomach” This sentence is not correct. The authors might refer to Image guided radiotherapy (IGRT), adaptive radiotherapy and/or 4D CT for radiotherapy planning. These techniques enable to take into account the mobility of stomach. On the contrary, IMRT does not take into account this mobility but enables to improve radiotherapy dosimetry with better planning target coverage and organs at risk sparing.

Response:

Thanks for your comments. We have revised the statements, and cited relevant articles accordingly in the Toxicity, therapeutical considerations, and biomarkers of NCRT section (Page 14, line 19-26).

Reviewer 2 Report

We read with interest this review regarding neoadjuvant chemoradiotherapy in gastric cancer.

Some changes are required before eventual publication.

There are some grammar mistakes and oversights throughout the manuscript that should be corrected. We recommend a linguistic professional service revision in order to help the readability of the paper.

Second, the background of medical treatment for gastric cancer patients should be better explained in the introduction, and some recently published paper included (PMID: 33916206 ; PMID: 33916915; PMID: 31793342).

Third, the current paper is not suitable for publication if the authors do not discuss the practice-changing FLOT trial. This study revolutionized treatment scenario in this setting and severely limited the role of chemoradiotherapy. The authors should discuss this point and add a specific paragraph.

Author Response

We appreciate your diligent review and precious comments to improve this article. Our responses and revision are presented below according to the comments:

  1. There are some grammar mistakes and oversights throughout the manuscript that should be corrected. We recommend a linguistic professional service revision in order to help the readability of the paper.

Response:

We highly appreciate your comments. We have made the necessary English editing in the revised version and hope the improvement will meet the journal’s requirement. The certificate is uploaded as an attachment file.

  1. The background of medical treatment for gastric cancer patients should be better explained in the introduction, and some recently published paper included (PMID: 33916206 ; PMID: 33916915; PMID: 31793342).

Response:

Thanks for your recommendation. We have briefly reviewed the development and progress of medical treatment for gastric cancer patients in the Introduction (Page 5, line 8-10), and the above articles are cited accordingly.  

  1. The current paper is not suitable for publication if the authors do not discuss the practice-changing FLOT trial. This study revolutionized treatment scenario in this setting and severely limited the role of chemoradiotherapy. The authors should discuss this point and add a specific paragraph.

Response:

Thanks for your recommendation in this point. We have discussed the FLOT-4 trial in the main article with a separated paragraph, under “NCRT versus NCT for EGJ and gastric cardia cancers” section (Page 9, line 6-14).

Round 2

Reviewer 1 Report

The authors provided a revised manuscipt which adressed all the suggested revisions

Reviewer 2 Report

The authors addressed all the queries and issues we raised.